# Evaluation of Photobiomodulation and Boldine as Alternative Treatment Options in Two Diabetic Retinopathy Models

**DOI:** 10.3390/ijms24097918

**Published:** 2023-04-27

**Authors:** Víctor Calbiague García, Bárbara Cadiz, Pablo Herrera, Alejandra Díaz, Oliver Schmachtenberg

**Affiliations:** 1Ph. D. Program in Neuroscience, Facultad de Ciencias, Universidad de Valparaíso, Valparaíso 2360102, Chile; 2Centro Interdisciplinario de Neurociencias de Valparaíso (CINV), Facultad de Ciencias, Universidad de Valparaíso, Valparaíso 2360102, Chile; 3Instituto de Biología, Facultad de Ciencias, Universidad de Valparaíso, Valparaíso 2360102, Chile

**Keywords:** photobiomodulation, retina, organotypic retinal explants, photoreceptors, diabetes, glucose, lactate

## Abstract

Diabetic retinopathy causes progressive and irreversible damage to the retina through activation of inflammatory processes, overproduction of oxidative species, and glial reactivity, leading to changes in neuronal function and finally ischemia, edema, and hemorrhages. Current treatments are invasive and mostly applied at advanced stages, stressing the need for alternatives. To this end, we tested two unconventional and potentially complementary non-invasive treatment options: Photobiomodulation, the stimulation with near-infrared light, has shown promising results in ameliorating retinal pathologies and insults in several studies but remains controversial. Boldine, on the other hand, is a potent natural antioxidant and potentially useful to prevent free radical-induced oxidative stress. To establish a baseline, we first evaluated the effects of diabetic conditions on the retina with immunofluorescence, histological, and ultrastructural analysis in two diabetes model systems, obese LepR^db/db^ mice and organotypic retinal explants, and then tested the potential benefits of photobiomodulation and boldine treatment in vitro on retinal explants subjected to high glucose concentrations, mimicking diabetic conditions. Our results suggest that the principal subcellular structures affected by these conditions were mitochondria in the inner segment of photoreceptors, which displayed morphological changes in both model systems. In retinal explants, lactate metabolism, assayed as an indicator of mitochondrial function, was altered, and decreased photoreceptor viability was observed, presumably as a consequence of increased oxidative-nitrosative stress. The latter was reduced by boldine treatment in vitro, while photobiomodulation improved mitochondrial metabolism but was insufficient to prevent retinal structural damage caused by high glucose. These results warrant further research into alternative and complementary treatment options for diabetic retinopathy.

## 1. Introduction

One of the most debilitating complications of diabetes is diabetic retinopathy (DR), which affects about one-third of all diabetics [1]. In uncontrolled diabetes, DR often proceeds to near-complete blindness through macular edema, vitreal hemorrhage, retinal traction detachment, ischemia, or retinal hemorrhages, and even under strict blood glucose and ophthalmologic control, severe visual impairment is a common consequence of DR [2]. DR has traditionally been diagnosed and staged according to an eye fundus evaluation, which reveals characteristic alterations in the retinal blood supply [3]. In early, non-proliferative DR, the fundus image displays microaneurysms, unperfused capillaries, and minor hemorrhages, while new but pathological blood vessels sprout in the retina and into the vitreous during later stages of proliferative DR. However, before vascular alterations become visible, sensitive methods such as focal electroretinography (ERG) reveal initial functional alterations in the diabetic retina, notably a significant reduction in ERG component amplitudes (a and b-waves; oscillatory potentials) and delays of their response peaks (“implicit time”) [4,5,6,7], which indicates early and possibly direct deleterious effects of diabetes on the neuroretina, notably photoreceptors.

Current treatment options for DR are primarily directed at vascular alterations and are mostly applied at advanced DR stages. These treatments are quite invasive and involve the periodic intravitreal injection of drugs, for example, the anti-VEGF medication bevacizumab. Therefore, there is a need for alternative or complementary, non-invasive treatment options that could be applied early on in the disease to reverse its course or delay its progression.

Photobiomodulation (PBM), the stimulation with generally near-infrared light at comparatively low intensities, has been applied for half a century to many different tissues and in a large array of pathologies [8,9,10,11]. PBM is especially attractive for the treatment of retinal pathologies since the retina is difficult to target systemically due to the blood-retina barrier, but readily accessible to photons. Previously, PBM treatment has produced mixed results in a variety of retinal pathologies and insults [12,13,14,15]. For instance, PBM ameliorated damage caused by bright light, blue light, and oxygen toxicity in rodents [12,13,15]. Moderately positive effects have been shown in one study of retinopathy of prematurity (ROP), while another one reported no real benefits [16,17]. Two recent studies in patients with age-related macular degeneration (AMD) also generated conflicting results [18,19].

In DR and its animal models, PBM has been tested in a few studies, which have mostly reported moderately positive effects on visual function, cell death rate, inflammation, oxidative stress, and retinal capillary permeability without harmful side-effects of the treatment. Several of these studies stem from the laboratory of T.S. Kern at Case Western Reserve University. In their first respective paper, rats were rendered type-I diabetic with streptozotocin and treated once daily for four minutes with 670 nm near-infrared light [20]. Markers for cell death, inflammation, oxidative stress, and vascular pathology were assayed and generally found to improve with PBM treatment. Two important caveats in this study are that the onset of diabetic retinopathy was not verified before PBM application and that streptozotocin is by itself retinotoxic, directly damaging photoreceptors in organotypic retinal explants [21].

A subsequent study applied the same stimulus protocol to streptozotocin-treated C57Bl/6J mice and evaluated, among others, retinal superoxide generation and leukostasis of the retinal vasculature [22]. Both markers for retinal pathology were reduced under PBM treatment, as were the expression levels of ICAM-1 and iNOS. Although measurements of retinal function (contrast sensitivity) are mentioned, they are not shown. A third and more comprehensive study applied the streptozotocin treatment protocol again to C57Bl/6J mice for a period of 8 months to test if long-term PBM could prevent the early neuronal and vascular lesions observed in diabetic retinopathy [23]. Visual function, retinal capillary permeability, and capillary degeneration were assayed. PBM was shown to prevent a diabetes-related degeneration of contrast sensitivity and spatial frequency thresholds and exert protective effects on the integrity of the retinal vasculature.

More recently, two clinical trials have evaluated PBM treatment for diabetic macular edema (DME). The first trial applied 670 nm laser light of up to 200 mW/cm^2^ to 21 DME patients [24]. The study reports the safety of the treatment and a significant reduction in central macular thickness in three patient groups after 2 and 6 months. However, the study did not involve any untreated or placebo control groups. A second clinical trial employed a larger number of DME patients, who were randomly assigned to 670 nm LED PBM or placebo groups [25]. While the treatment did not cause adverse reactions, no significant effects of PBM on foveal thickness or visual acuity were observed during the study’s duration of 4 months. Finally, a systematic review of nine studies and clinical trials on PBM application for DR, including some of the studies mentioned above, cautiously concluded that PBM therapy might “fill a role in treating niches of DR and AMD patients”, once a consensus on treatment parameters has been reached [26].

Clearly, PBM still holds promise, but more basic research is needed before the treatment should be applied in a clinical setting. Here, we set out to comprehensively evaluate the degree of damage to the neural retina in two complementary diabetes model systems, each offering unique advantages: Organotypic retinal explants from wild-type C57BL/6J mice and obese/diabetic LepR^db/db^ mice (db/db). This mouse model has previously been characterized as useful for the study of diabetic retinopathy, displaying progressive cell loss through apoptosis and glial activation [27]. Finally, we evaluated the potential benefits of PBM treatment in retinal explant culture subjected to high glucose concentrations to emulate diabetic conditions, and tested the effects of the potent antioxidant boldine, an alkaloid of the Chilean Boldo tree (*Peumus boldus*), as an alternative treatment option to reduce oxidative stress [28].

## 2. Results

### 2.1. Retinal Cell Survival in a Diabetic Model Mouse

In order to establish the baseline of the effects of diabetic condition on the retina, we analyzed retinal morphology and used immunofluorescence to visualize the expression of cell type-specific markers to determine the state of the five neuronal cell types of the ex vivo retina in wild-type and db/db mice at 2–3 (young-adult; *n* = 8 animals) and 6–8 months of age (old-adult; *n* = 8 animals) (Figure 1). First, we evaluated the preservation of photoreceptors by measuring the expression of S-arrestin, a specific marker for rod photoreceptors (rods) [29]. Interestingly, we did not observe any alteration in the number of rods, either in young-adult (control = 193.1 ± 30.4, *n* = 4; db/db = 160.6 ± 27.4, *n* = 4) or in old-adult animals (control = 184.7 ± 10.7, *n* = 4; db/db = 165.0 ± 27.6, *n* = 4) (Figure 1A,B). Furthermore, in young-adult db/db mice, no morphological alteration was observed in the outer retinas (OS length = 9.4 ± 1.7 µm, *n* = 4; IS length = 9.9 ± 0.8 µm, *n* = 4) versus control mice (OS = 13.9 ± 3.7 µm, *n* = 4; IS = 11.1 ± 1.9 µm, *n* = 4) (Figure 1D,E). Yet, a reduction in the length of the IS was evident in old-adult diabetic retinas compared to controls (IS db/db = 9.8 ± 1.1 µm, *n* = 4; IS control = 12.2 ± 0.9 µm, *n* = 4, *p* = 0.014) (Figure 1E). The OS remained unaltered in old-adult db/db mice (OS control = 13.7 ± 2.9 µm, *n* = 4, OS db/db = 10.1 ± 2 µm, *n* = 4) in comparison to controls (Figure 1D,E).

Consequently, we focused on the thickness of the different retinal cell layers. A difference to wild-type mice was observed in the thickness of retinas of old-adult db/db mice, but not in young-adult db/db mice (Figure 1G). The average thickness of the retina for young-adult mice was 191.7 ± 6.2 µm and 172.4 ± 25 µm for control (*n* = 5) and db/db mice (*n* = 4), respectively. In old-adult mice, the thickness of control (*n* = 3) and db/db (*n* = 4) mice were 194.5 ± 4.8 µm and 170.1 ± 5.8 µm, respectively (*p* = 0.0410) (Figure 1G).

Accordingly, in young-adult animals, ONL (control = 55.2 ± 6.3 µm, *n* = 4; db/db = 58.9 ± 2.3 µm, *n* = 4), OPL (control = 9.1 ± 0.9 µm, *n* = 5; db/db = 8.2 ± 2.1 µm, *n* = 4), INL (control = 9.1 ± 0.9 µm, *n* = 5; db/db = 8.2 ± 2.2 µm, *n* = 4), and IPL (control = 46.6 ± 2.9 µm, *n* = 5; db/db = 40.1 ± 7.2 µm, *n* = 4) did not display thickness alterations compared to wild-type controls, while there was a reduction in the GCL (control = 20.6 ± 1.1 µm, *n* = 5; db/db = 13.4 ± 1.5 µm, *n* = 4, *p* = 0.0017) (Figure 1C,H–K).

Interestingly, in old-adult animals, neither individual layer, ONL (control = 57.9 ± 3.2 µm, *n* = 3; db/db = 55.8 ± 2.7 µm, *n* = 4), OPL (control = 8.7 ± 1.9 µm, *n* = 3; db/db = 9.2 ± 0.5 µm, *n* = 4), INL (control = 8.7 ± 1.9 µm, *n* = 3; db/db = 9.2 ± 0.5 µm, *n* = 4), IPL (control = 41.8 ± 2.1 µm, *n* = 3; db/db = 38.1 ± 4.1 µm, *n* = 4) or GCL (control = 15.3 ± 0.8 µm, *n* = 3; db/db = 15.1 ± 2.3 µm, *n* = 4) displayed thickness reductions compared to wild-type controls, while their sum generated the significant difference mentioned above (Figure 1C,H–K).

To determine cell survival under diabetic conditions in the inner retina, we used markers for horizontal cells (HC; calbindin), rod bipolar cells (RBC; PKCα), amacrine cells (AC; calretinin), and retinal ganglion cells (RGC; β- III-Tubulin) (Figure 1F). However, the density of the different inner retinal cell types did not display changes, neither in young-adult nor in old-adult db/db mice (Figure 1L–N).

Taken together, these results indicate that gross retinal morphology and cellular survival in the inner retina of young-adult and old-adult mice are largely maintained, and retinal thickness is only slightly decreased in db/db mice. The inner segments of the photoreceptors show a reduction, contributing to the slight overall thinning of the diabetic retinas.

### 2.2. Diabetic Conditions Produce Morphological Changes of Mitochondria in the Inner Segment of Photoreceptors

A reduction in rod inner segment length may be indicative of altered mitochondrial density and/or morphology and thus a possible effect on energy metabolism, which is crucial for photoreceptor function due to their exceptionally high energy demands [30]. Since mitochondria are highly plastic and display both fission and fusion, we set out to test the idea of a possible effect of diabetic conditions on retinal mitochondria and studied the morphology of this organelle in the photoreceptor inner segments by transmission electron microscopy.

First, we evaluated the mitochondria in wild-type control (3 animals) and diabetic db/db mice (3 animals). A subtle difference was observed at the ultrastructural level between both animal models (Figure 2). Mitochondria tend to have an elongated morphology, and they are capable of fusing to form a mitochondrial network in cells [31]. In control animals, we observed the classic elongated morphology of the mitochondria and their cristae. Strikingly, diabetic mice displayed a reduced size and a more rounded shape, compared to control mice (Figure 2A,B). To quantify this observation, we measured the mitochondrial length-to-width ratio. This analysis reveals that in diabetic mice, the mitochondria do have a more rounded morphology (4.4 ± 3.5; *N* = 3 animals, *n* = 138 mitochondria; *p* < 0.0001) (Figure 2A), compared to those of healthy control mice (6.5 ± 3.4; *N* = 3 animals, *n* = 135 mitochondria).

Previously, we had shown that organotypic retinal explant cultures are a good tool to assess the effects of high glucose conditions on the neuro-retina [32,33]. Therefore, to study if a high glucose condition in explant culture causes similar morphological alterations in mitochondria as those observed in diabetic mice, we compared these organelles in photoreceptor IS in explants cultured under 15 mM (moderate glucose, *N* = 3 explants) versus 30 mM glucose (high glucose, *N* = 3 explants). Indeed, we observed a similar rounding effect under these in vitro hyperglycemic conditions (Figure 2B). Even when exposed to moderate glucose concentrations, the mitochondria exhibited a rounded morphology compared to control mice (2.0 ± 1.1, *N* = 3 explants; *n* = 626 mitochondria), but the effect was exacerbated under high glucose conditions, where the length/width ratio was close to 1 (1.5 ± 0.5, *N* = 3 explants; *n* = 606 mitochondria; *p* < 0.0001).

Altogether, these results demonstrate the effects of diabetic conditions in vivo and in vitro on mitochondria in photoreceptor inner segments and may reflect an altered energy metabolism.

### 2.3. Retinal Cell Death Rate Is Increased by Exposure to High Glucose and Nitrosative Stress

Many studies have proposed that diabetic conditions disrupt mitochondrial functions, leading to increases in oxidative and nitrosative stress [34,35,36], where the principal oxidative agents are hydrogen peroxide, the hydroxyl radical, NO, the superoxide anion, and peroxynitrite, the latter likely being the most harmful [35]. To study the role of nitrosative stress in the cell death process in retinal cells under high glucose conditions, we treated a total of 42 retinal explants (21 wild-type animals) with SIN-1 for up to 2 weeks in culture and measured the cell death rate after 7 and 14 days in culture. SIN-1 is a molecule that spontaneously releases nitric oxide and superoxide anions, which rapidly react to form peroxynitrite under physiological conditions. It is therefore considered a peroxynitrite donor [37].

The TUNEL assay confirmed an elevated cell death rate after 7 days in culture, in the ONL under moderate glucose conditions + SIN-1 compared to control conditions (15 mM glucose) (Figure 3A,B); (340.2 ± 97.2%; *n* = 3 explants, *p* = 0.0275). This cell death rate was similar to the rate measured in retinal explant cultures under high glucose conditions (386.2 ± 77%; *n* = 3 explants, *p* = 0.0016). It was also similarly elevated after 14 days for the moderate glucose + SIN-1 (367.2 ± 124.5%; *n* = 3 explants, *p* = 0.0077) and high glucose conditions (378.2 ± 70.5%; *n* = 3 explants, *p* = 0.0001) (Figure 3B). Note that for this and the following results, the data are normalized versus control explants (dashed line at 100%) and the *p*-value indicates significance compared to controls.

Since the high glucose condition caused a cell death rate comparable to that of control explants treated with a peroxynitrite donor, we set out to test if endogenous NO synthesis was involved in the elevated cell death rates. To this end, we treated retinal explants with L-NAME, a general inhibitor of the three isoforms of nitric oxide synthetase (NOS). The addition of this molecule to our culture conditions for up to 7 days decreased the cell death rate in the high glucose condition to levels not different from controls (100.0 ± 66.5%; *n* = 3 explants, *p* > 0.999), supporting a relationship between upregulation of NOS and cell death (Figure 3A,B). This preventive effect was preserved after 14 days in culture (75.86 ± 16.6%; *n* = 3 explants, *p* = 0.8685) (Figure 3B). These results suggest that the increment in the photoreceptor cell death rate under high glucose conditions is related to NO overproduction and resulting nitrosative stress.

Given the involvement of oxidative/nitrosative stress in photoreceptor cell death under diabetic conditions, we set out to evaluate antioxidants as preventive agents of the nitrosative effects on the retina under high glucose conditions. To this end, we treated retinal explant cultures with boldine, which is known for its antioxidant properties, apart from its reported role as a hemichannel blocker [38,39]. To establish if boldine could counteract the effects caused by nitrosative stress, we treated retinal explants with SIN-1 together with boldine (Figure 3A). Strikingly, treatment with boldine prevented the increment in the cell death rate produced by SIN-1 after 7 days (140.2 ± 75.9%; *n* = 3 explants, *p* = 0.9267) and 14 days (58.6 ± 61.2%; *n* = 3 explants, *p* = 0.8899), suggesting that boldine neutralized the excess of reactive nitrosative species. In addition, when retinal explants under high glucose conditions were co-administered boldine, the increment in TUNEL positive photoreceptors was also prevented, both after 7 days (127.6 ± 35%; *n* = 3 explants, *p* = 0.9377) and 14 days in culture (29.3 ± 7.9%; *n* = 3 explants, *p* = 0.1930) (Figure 3B).

To further confirm that a high glucose condition generates a nitrosative environment in the retina, we labeled the cultured retinal explants (the same animals and explants used to study the cell death rate, *n* = 42 explants from *N* = 21 animals) with an antibody against nitrotyrosine (N-tyr), which is a product of tyrosine nitration mediated by reactive nitrogen species such as peroxynitrite anion and nitrogen dioxide and therefore considered a marker of NO-dependent oxidative stress [40].

Indeed, retinal explant slices showed immunolabelling throughout the retina (Figure 3C). Control conditions cultured for either 7 days or 14 days displayed comparatively low levels of immunolabelling. As expected, N-tyr levels were incremented by the peroxynitrate donor SIN-1 in moderate glucose conditions after 7 days (154.9 ± 5.4%; *n* = 3 explants *p* < 0.0001) and 14 days (202.4 ± 49.2%; *n* = 4 explants, *p* = 0.0342) (Figure 3D), with N-tyr levels comparable to those observed in retinal explants cultured under high glucose after 7 days (132.9 ± 5.2%; *n* = 3 explants, *p* = 0.0223) or 14 days (186 ± 72.4%; *n* = 4 explants, *p* = 0.1934) (Figure 3D). Interestingly, the treatment of retinal explants cultured under high glucose conditions with the NOS inhibitor L-NAME decreased the nitrosative stress produced by 30 mM glucose to values exhibited by the control explants under moderate glucose after 7 days (67.8 ± 19.8%; *n* = 3 explants, *p* = 0.0252) or 14 days (103.0 ± 41.6%; *n* = 4 explants, *p* = 0.9998). This effect was also observed after a treatment with boldine in retinal explants cultured under high glucose conditions for up to 7 days (78.1 ± 4.7%; *n* = 3 explants, *p* = 0.1312), and 14 days (91.7 ± 58%; *n* = 4 explants, *p* = 0.9967) (Figure 3D).

In summary, these results support the notion that high glucose conditions cause an increase in the nitrosative stress levels in the retina, which elevate the cell death rate mainly of photoreceptors. This outcome can be prevented by inhibition of NOS and by antioxidant agents such as boldine, suggesting a pathological upregulation of NO synthesis as a crucial factor in high glucose-induced photoreceptor cell death.

### 2.4. Prolonged Exposure to High Glucose Leads to an Inflammatory Environment and Increased Nitric Oxide from iNOS in Retinal Explants

Having confirmed the increase in nitrosative stress in retinal explants cultured under high glucose after 7 and 14 days in culture, we investigated the source of NO that may produce the overproduction of peroxynitrate that may underlie the observed nitrosative levels. Therefore, using the same number of explants and animals used in the previous section (*n* = 42 explants, *N* = 21 wild-type animals), we studied how the 30 mM glucose condition affected the expression of two NOS isoforms: neuronal NOS (nNOS) and inducible NOS (iNOS) [41]. Since organotypic retinal explants lack vascularization, we decided not to investigate endothelial NOS (eNOS), as this isoform is mainly expressed in endothelial cells of blood vessels [42]. Immunolabelling experiments to study the expression of nNOS showed the presence of nNOS-positive amacrine cells, termed NOACs, and their processes in the IPL (Figure 3E; [41]), in all conditions. No differences in the number of nNOS-positive cells were observed between the conditions, indicating that the expression of this NOS isoform is not affected by prolonged high glucose treatment (Figure 3F).

On the other hand, iNOS, the NOS isoform related to inflammatory processes [43], displayed widespread diffuse labeling throughout the retina, with higher overall expression in the outer and inner plexiform layers and in photoreceptor inner segments (Figure 3G). In retinal explants under control conditions, an increase in the fluorescence intensity was observed under high glucose conditions either after 7 days (224.2 ± 111.9%; *n* = 3 explants, *p* = 0.1852) or 14 days in culture (302.3 ± 94%; *n* = 4 explants, *p* = 0.0067). Interestingly, SIN-1 addition also caused an elevation of iNOS expression throughout the retina at 7 days (236.2 ± 72.8%; *n* = 3 explants, *p* = 0.1420) and 14 days in culture (313.0 ± 132.7%; *n* = 4 explants, *p* = 0.0141), which might be related to the concomitant increase in neuroinflammatory conditions with nitrosative stress (Figure 3H).

Remarkably, co-treatment with boldine abolished the effects of SIN-1 in control conditions after 7 days (115.7 ± 70.4%; *n* = 3 explants, *p* = 0.9915) and 14 days (109.5 ± 52.3%; *n* = 4 explants, *p* = 0.9983). The treatment with boldine had the same effect under high glucose conditions after 7 days (115.7 ± 48.3%; *n* = 3 explants, *p* = 0.9911) and 14 days (130.4 ± 60.2%; *n* = 4 explants, *p* = 0.9239) (Figure 3G,H). As boldine, L-NAME also reduced iNOS expression in retinal explants under high glucose conditions either at 7 days (98.1 ± 22.4%; *n* = 3 explants, *p* > 0.9999) or 14 days in culture (158.3 ± 60.8%; *n* = 4 explants, *p* = 0.6463), which may again reflect positive feedback between nitrosative stress and neuroinflammatory conditions. Altogether, these results indicate that the main source of NO in retinal explants under high glucose is iNOS, leading to overproduction of peroxynitrite in this pathological condition.

### 2.5. Effect of Photobiomodulation Treatment on Retinal Structure and Metabolism under High Glucose Conditions

Since the retina is readily accessible to photons, we tested photobiomodulation (PBM) as a possible treatment option to diminish or reverse the deleterious effects of high glucose-induced oxidative/nitrosative stress [44].

To study possible functional effects of PBM, we treated a total of 24 retinal explant cultures (12 wild-type animals) for up to 7 days in a black box, with either of two LED sources inside emitting at two different wavelengths: 525 nm (control) and 660 nm (near infrared, NIR). We first investigated how PBM affects the gross structure of organotypic retinal explants under moderate (15 mM; 525 nm, *n* = 3; 660 nm, *n* = 3) and high (30 mM; 525 nm, *n* = 3; 660 nm, *n* = 3) glucose concentrations. PBM treatment with NIR light maintained the overall retinal thickness under moderate glucose conditions (171.9 ± 34.9 µm; Figure 4A,B; *p* = 0.0261) compared to green light stimulation (99.1 ± 11.1 µm). This was mainly due to an effect on the inner retina, with a decrease in IPL thickness of 10.2 ± 9.2 µm (Figure 4A,B; 525 nm 15 mM = 15.2 ± 2.1 µm; 660 nm 15 mM = 27.8 ± 8.9 µm; *p* = 0.0769) and of INL thickness of 29.7 ± 31.2 µm (Figure 4A,B; 525 nm 15 mM = 25.8 ± 2.3 µm; 660 nm 15 mM = 55.6 ± 31.1 µm; *p* = 0.1742), and a decrease of OPL thickness of 4.6 ± 1.7 µm (Figure 4A,B; 525 nm 15 mM = 8.8 ± 0.9 µm; 660 nm 15 mM = 13.4 ± 1.5 µm; *p* = 0.011). However, under high glucose, no difference between NIR and green light stimulation could be observed, either in the complete retina or its sublayers (Figure 4C).

Because it is generally thought that PBM affects mitochondrial electron transport, enhancing ATP production [45,46], we proceeded to investigate retinal metabolism after PBM modulation, focusing on Müller cells (MCs). We decided to study this cell type due to its central role in metabolic coupling, energy flux, and support to the neuroretina [47,48,49].

To that end, we expressed the lactate nanosensor Laconic in 28 retinal explant cultures (14 animals) treated with PBM under different glucose conditions (Figure 5A). Recently, we have shown the feasibility of this assay in MCs [50]. First, we determined if PBM changes the fluorescence properties of the nanosensor. For this, we calculated the delta ratio (ΔR) of the Laconic by depleting intracellular lactate levels with 10 mM pyruvate, and saturating lactate levels with 10 mM lactate. To deplete intracellular levels of these metabolites, we used a property of monocarboxylate transporters (MCTs) called trans-acceleration [51], where an extracellular exposure to a monocarboxylate triggers intracellular substrate efflux. The ΔR for Laconic for each PBM stimulation and glucose condition was: 525 nm at 15 mM (control, N = 4 explants, n = 7 cells) = 23.5%, 525 nm at 30 mM = 24.3% (*N* = 3, *n* = 9), 660 nm at 15 mM (control, *N* = 3, *n* =5) = 22.5%; 660 nm at 30 mM = 23.4% (*N* = 3, *n* = 7) (Figure 5B). These results indicate that PBM does not affect the fluorescence properties of the nanosensor.

Once the functional expression of Laconic was confirmed after PBM stimulation, we used the metabolic sensor to study intracellular lactate metabolism, using the so-called transport-stop protocol [52,53]. This protocol is based on the inhibition of MCTs, allowing to isolate intracellular lactate synthesis and consumption. To inhibit the different MCTs isoforms expressed in the retina, we used a cocktail of 3 different drugs (SR. 13800, ARC-155858, and syrosingopine). Under basal culture conditions, the application of this pharmacology cocktail via bath perfusion produced an increase in intracellular lactate levels, under 525 nm and NIR wavelengths, in both glucose concentrations (15 mM and 30 mM), supporting a lactate producer profile in MCs (Figure 5C,D).

However, the retinal explants under the high glucose condition stimulated with 525 nm (0.02 ± 0.008, *N* = 3, *n* = 9) displayed a higher lactate production rate in comparison with the moderate glucose condition under 525 nm (0.008 ± 0.004, *N* = 3, *n* = 11, *p* = 0.0385). This alteration was abolished in both glucose conditions treated with NIR (Figure 5E), which showed a higher lactate production compared to the retinal explants treated with 525 nm at 15 mM (660 nm 15 mM = 0.04 ± 0.01, *N* = 3, *n* = 5, *p* = 0.0004; 660 nm 30 mM = 0.03 ± 0.01, *N* = 3, *n* = 11, *p* = 0.0007) (Figure 5E).

Next, we evaluated how lactate dynamics change under different PBM conditions. Interestingly, under stimulation at 525 nm and 660 nm, when the retina was subjected to general depolarization with 12 mM KCl, lactate metabolism changed, and MCs started to consume lactate under both high glucose and control conditions (Figure 5C–F). Taken together, these results suggest that retinal explants treated with 525 and 660 nm display a high lactate consumption. Because lactate is converted into pyruvate to be metabolized in the Krebs cycle, these results support the notion that retinas under NIR stimulation exhibit an increased mitochondrial metabolism.

Altogether, these results suggest that PBM affects retinal metabolism, producing an enhancement in mitochondrial oxidation, but that this effect is insufficient to improve retinal structural damage caused by high glucose conditions in retinal explant cultures.

## 3. Discussion

In our study, we were able to demonstrate that exposure to high glucose caused changes in the mitochondrial morphology within photoreceptor inner segments and triggered significantly higher rates of cell death in photoreceptors, which was related to an increase in nitrosative stress. Treatment with boldine was capable of preventing alterations caused by high glucose. In addition, retinal explants treated with PBM displayed an improvement in mitochondrial oxidation, but this effect was insufficient to improve retinal structural damage related to high glucose conditions.

There is a need for complementary, non-invasive treatment options for diabetic retinopathy and other retinal pathologies that could be applied early on in disease progression to decelerate or reverse its course. Due to the very high energy demand of photoreceptors and their sensitivity to oxidative/nitrosative stress, mitochondria might be an innovative treatment target, and PBM is an obvious candidate [44]. PBM is evidently most suitable for tissues accessible to light stimulation, such as skin, mucosa, and the retina, but its more recent experimental applications also include inner organs, bones, the cochlea, and the brain [11,54,55,56,57]. Near-infrared light is able to penetrate deep into tissues, as it is much less absorbed by tissue and hemoglobin than shorter wavelength light [58]. Generally, transiently increased cellular metabolism and ATP synthesis are observed after PBM application [45], and it has been established that mitochondrial enzymes are sensitive to near-infrared light stimulation [59,60,61,62,63]. Mitochondrial cytochrome c oxidase (CCO), or complex IV, is generally considered the main target of PBM [64,65].

Several theories have been proposed over the years regarding how PBM affects mitochondrial electron transport and CCO activity. One theory states that photic excitation of the metal centers of CCO causes an increase in electron transfer speed in the mitochondrial electron chain [46,66], leading to higher respiratory efficiency, which would reduce the generation of ROS and oxidative stress, due to fewer “lost” electrons generating superoxide [67,68,69].

Another hypothesis proposes that PBM disinhibits CCO by promoting the dissociation of inhibitory bound NO from the enzyme [65,70]. Although the NO concentrations required for CCO inhibition are beyond the physiological range of NO signaling [71], this effect could be relevant under pathological or inflammatory conditions. CCO is a complex protein composed of two heme centers (heme a and a3), two active copper sites (Cu A and Cu B), a zinc center, and a magnesium center. Of these, four are considered redox centers (CuA, CuB, heme a, and heme a3). CCO was shown to have overlapping light absorption and action spectra in the 420–450 nm, 620–680 nm, and 760–830 nm ranges, supporting the stimulation of its copper centers by PBM [60,61]. Accordingly, PBM increases the oxygen consumption of isolated hepatic mitochondria and the mitochondrial energy charge in muscle cells [72]. In cardiac and hepatic tissue, it was shown to raise the production of ATP [73,74,75]. Several authors have reported an increase in cell proliferation after PBM application, among others, in human fibroblasts [76], adipose-derived stem cells [77], mouse primary hepatocytes [78], an osteoblast precursor cell line (MC3T3-E1) [79], and in neurons of the post-ischemic brain [80]. It should be mentioned that CCO stimulation is not universally accepted as the core mechanism of PBM [81], and alternative mechanisms have been proposed, among others, that PBM acts as an activator of ROS signaling pathways [78,82].

Photoreceptors are estimated to contain at least 75% of all retinal mitochondria [83], generating a high basal load of free radicals. Under physiological conditions, oxidative species can be rapidly neutralized by enzymatic mechanisms; however, in DR, higher free radical production is accompanied by a decrease in antioxidant enzyme expression, causing an oxidant/antioxidant imbalance [84]. Yu et al., 2006 [85] demonstrated that under high glucose conditions, reactive oxygen species increase and generate a change in mitochondrial morphology in a heart muscle cell line, suggesting that elevated mitochondrial activity leads to an increase in reactive oxygen species (ROS), which consequently produces an irreversible imbalance in the fission/fusion dynamics of the mitochondrial network [85]. Here we were able to observe that under high glucose conditions and in diabetic animals, mitochondria displayed a tendency to be shortened and rounded, suggesting that these morphological changes could reflect mitochondrial damage. While the mechanisms that could lead to mitochondrial dysfunction are not yet well understood, it is believed that CCO inhibition could activate a variety of effects, from inhibition of mitochondrial oxidative phosphorylation to ROS generation and apoptosis [35].

Diabetes-induced neuronal death in the retina has been documented previously [32,33,86]. There is evidence that before photoreceptor death is observed, a series of morphological changes occur, such as degeneration of photoreceptor outer segments and changes in opsin distribution [87]. According to the results obtained here, photoreceptors are the most affected neurons under diabetic conditions, which suggests that exposure to elevated glucose levels causes molecular alterations eventually leading to apoptosis and diminished retinal function.

Our results support an idea that has gained traction in recent years: Photoreceptors could be major contributors to oxidative stress and local inflammation in diabetic retinopathy [88,89]. While the mechanisms underlying this hypothesis are still a matter of discussion, we demonstrate by immunohistochemistry that under high glucose conditions in retinal explants, 3-nitrotyrosine labeling is significantly increased. There is evidence that incubation of various cell types with exogenous peroxynitrite induces apoptosis, which would indicate that peroxynitrite might be able to activate specific intracellular signaling pathways to stimulate apoptosis instead of just causing nonspecific molecular oxidation [90]. Furthermore, peroxynitrite overproduction has been linked to the cause of several inflammatory pathologies and autoimmune and neurodegenerative diseases [91,92], as it is able to oxidize and damage not only proteins but also lipids and DNA in a nonspecific and irreversible manner. Examples include the inactivation of heat shock protein 90 (Hsp90) [90,92], activation of poly-ADP-ribose polymerase-1 [56], and damage to manganese superoxide dismutase, with the consequence of free radical accumulation leading to permanent damage to mitochondrial metabolism [90].

The increased presence of nitrotyrosine and iNOS under high glucose and SIN-1 treatment demonstrate that neuroinflammation develops in parallel with oxidative stress in the retina, which could explain eventual cell death in photoreceptors. Studies with knockout animals for iNOS have shown that this enzyme is an important component in the early inflammatory state in diabetic animals and that its expression is key to inducing nitrosative stress and protein modification through the action of peroxynitrite [93]. Other studies support this idea, demonstrating that upregulated retinal iNOS favors an inflammatory microenvironment and overproduction of NO [94], and can induce the production of other inflammatory proteins such as ICAM-1 in photoreceptors [83]. In addition, increased mitochondrial activity implies an elevated release of the superoxide anion, which rapidly reacts with NO to form peroxynitrite, the most damaging of the reactive nitrogen species [95,96,97]. 

The search for alternative and effective therapies strives to prevent or reduce the cell damage produced by oxidative/nitrosative stress in the retina. Therefore, in our study, we applied boldine, an alkaloid from the Chilean *Peumus boldus* tree that has been described to have antioxidant and anti-inflammatory properties [28,98]. We also studied the effects of L-NAME, a non-selective NOS inhibitor. The results were surprising, since in both cases, we were able to largely prevent the consequences of high glucose and the peroxynitrite donor SIN-1. Boldine has been used in both in vitro and in vivo models. In the db/db mouse, a model of obesity and diabetes, oral administration of boldine restored aortic endothelial damage by reducing nitrotyrosine expression, suggesting that this alkaloid improved endothelial function by reducing ROS and exerting a protective effect [98]. Similar results have been obtained in streptozotocin-induced type 1 diabetic animals, where boldine reversed the increase in ROS in endothelial cells by inhibiting oxidative stress and restoring NO levels in the cells [99]. The study of mitochondrial function in streptozotocin-treated mice indicated that boldine is able to decompose ROS and also inhibits NO overproduction by attenuating nitration by peroxynitrite [100]. This antioxidant capacity of boldine could delay or prevent the development and progression of diabetes in different disease models, as observed in the present study.

As boldine, NOS inhibitors have been previously used as potential therapeutic molecules. In this study, the use of L-NAME was shown to not only prevent the overproduction of NO from iNOS but also reduce photoreceptor cell death under high glucose conditions. These results confirm the results obtained by other authors, who have shown that L-NAME administration is able to reduce reactive oxygen species and increase pancreatic secretory function in diabetic animals [101]. The use of isoform-specific NOS inhibitors might further improve outcomes, as it would allow inhibition of pathological NO overproduction without altering the physiological functions of nNOS, which in the retina is mainly synthesized by a type of amacrine cells. S-methylisothiourea sulfate and aminoguanidine, selective iNOS blockers, have been investigated in type II diabetes-induced neuropathic pain in Wistar strain rats and were shown to decrease pain through inhibition of iNOS and consequently attenuating mechanisms of NO-dependent neurodegeneration [95,102].

The main limitation of the study is the small sample size, which can limit the generalizability of the findings and make it difficult to draw conclusions because of the limited statistical power of our data. Another limitation is the model of retinal explants, which lack a functional vascular system. Furthermore, the comparatively short time that the tissue can be exposed to diabetic conditions—weeks compared to years or decades in vivo—does not allow to mimic the slow deleterious processes involved in chronic pathologies. Notwithstanding these caveats, rodent retinal explants have the advantage of allowing precisely controlled conditions and remain a useful tool for the study of the effects of diabetic retinopathy, allowing a direct and fast approach to testing potential complementary treatments, as demonstrated here, that may lay the groundwork for future research in in vivo models.

Altogether, the cited studies and the evidence presented here indicate that boldine and PBM are good candidates for complementary treatment options, where the main subcellular target is mitochondria, an organelle that is central to diabetic pathology (Figure 6).

## 4. Materials and Methods

### 4.1. Animals

A total of 61 wild-type C57Bl/6 (8 animals for immunofluorescence experiments, 3 animals for electron microscopy, and 50 animals for retinal explants) and a total of 11 obese/diabetic LepR^db/db^ mice (8 animals for immunofluorescence experiments, and 3 animals for electron microscopy) were housed under standard white cyclic illumination in a 12 h photoperiod with water and food ad libitum and were used irrespective of gender. Animals were born and raised in the animal facility at the University of Valparaiso. All efforts were made to minimize the number of animals used and their suffering. Protocols were approved by the bioethics committee of the University of Valparaiso and in accordance with Chilean animal protection law No. 20.380. To isolate the retinas, mice were deeply anesthetized with isoflurane (Baxter Healthcare Corporation, Guayama, Puerto Rico) before being sacrificed by decapitation.

### 4.2. Organotypic Retinal Explant Culture

A total of 97 retinal explants obtained from P9 wild-type mice from 50 wild-type animals were cultured as previously described [32,33] and divided among the experiments. Briefly, the retina was placed with the pigment epithelium facing down on cell culture inserts (Millicell, PIHA03050, Merck, Darmstadt, Germany) with 1 mL of culture containing 10% of fetal bovine serum (FBS), which was replaced every two days. The cultures were incubated at 37 °C in a 5% CO_2_ water-jacketed incubator (Thermo Scientific, Waltham, MA, USA). Once retinal explants were cultured, an adaptation period of two days was performed before the start of the different treatments. Moderate glucose concentration (15 mM) was used as a control, and high glucose (30 mM) was used to simulate diabetic conditions. The reagents used in this study were boldine (100 µM) (Cat# B3916-5, Sigma Aldrich, St. Louis, MO, USA), L-NAME (1 mM) (Cat# 80210, Cayman, Ann Harbor, MI, USA), and SIN-1 (10 µM) (Cat# 0756, Tocris Bioscience, Bristol, UK).

### 4.3. Immunostainings

The retinal slices were rehydrated with phosphate-buffered saline (PBS), followed by blocking for 1 h in a solution containing 1% bovine serum albumin (BSA), 1% horse serum, and 0.3% Triton X-100 (CAS# 9002-93-1, Amresco, Solon, OH, USA) in PBS. Glial cells were labeled with rabbit anti-glial fibrillary acidic protein (GFAP) 1:500 (Cat# 13-0300, RRID: AB_2532994, Thermo Fisher, Waltham, MA, USA). Primary antibodies were diluted in blocking solution and applied overnight at 4 °C. The sections were rinsed in PBS and incubated for 1 h at RT with the secondary antibody, goat anti-rabbit Alexa Fluor 488 1:1000 (Cat# A11034, RRID: AB_2576217, Thermo Fisher Scientific, Paisley, UK). All tissue sections were counterstained with DAPI and mounted in Fluoromount mounting medium (Fluoromount-G, Cat# 0100-01, Southern Biotech, Birmingham, AL, USA). The specificity of secondary antibodies was tested by omitting the primary antibody, which did not result in any labeling. Images were obtained with a confocal microscope (Nikon C1 Plus, Nikon Instruments, NY, USA).

### 4.4. TUNEL Assay

A terminal deoxynucleotidyl transferase dUTP nick end labeling (TUNEL) kit (Roche Diagnostics, Mannheim, Germany) was used for the detection of apoptotic cells. Cryosections of mouse retinal explants were cultured for 7 days, rehydrated with PBS, and permeabilized for 30 min with 0.3% Triton X-100 and 0.1% sodium citrate in PBS. The TUNEL reaction was performed for 1 h at 37 °C. The sections were rinsed in PBS, counterstained with DAPI, and mounted in Fluoromount mounting medium (Fluoromount-G, Cat# 0100-01, Southern Biotech, Birmingham, AL, USA). Confocal images were analyzed by counting TUNEL-positive nuclei for each nuclear layer of the retina over a 120 μm long stretch of the retina.

### 4.5. Transmission Electron Microscopy (TEM)

Mouse retinal explants were fixed in Karnovsky’s solution (2.5% glutaraldehyde in 0.1 M cacodylate buffer) for 1 h, washed in 0.1 M cacodylate buffer, immersed in reduced osmium tetroxide (EMS, Hatfield, PA, USA), dehydrated, and embedded in Epon resin (Embed-812, EMS, USA). Ultrathin sections of 70 nm thickness were mounted on copper grids and contrasted with uranyl acetate and lead citrate. Special care was taken to obtain identical transverse section angles between db/db and control mice for mitochondrial analysis. Ultrastructural images were obtained with a transmission electron microscope (Jeol JEM 1400 Flash) at 80 kV.

### 4.6. FRET Measurements

At p11 and p12, the explants were transduced by overnight incubation with 5 × 10^6^ plaque-forming units (PFU) of AAV-Laconic and imaged after two weeks in culture as previously described [50]. The AAV-GFAP-Laconic was constructed by the viral vector facility of ETH Zürich (Laconic: Addgene #44238).

Retinal slices were excited at 430 nm and visualized at 480 nm and 530 nm peak wavelengths, as previously reported [53]. All experiments were performed at room temperature (22–25 °C) with an upright fluorescence microscope (Olympus BX51) equipped with a 40× water-immersion objective, an Optosplit II emission image splitter (Cairn, Faversham, UK), and a Sensicam QE digital camera (Cooke Corp., MI, USA). Data acquisition was performed by custom software written in Python 4.0.1.

At the end of the experiments, data were exported for offline analysis of fluorescence intensities from each channel. To obtain the FRET ratio for Laconic (mTFP/Venus) fluorescence intensity values from each ROI and background were measured in ImageJ, version 1.52p (NIH, RRID: SCR_003070). In this study, all ROIs were chosen from the somas of MCs. The FRET data are displayed as the relative FRET ratio, expressed as the percentage of change over time of single experiments.

To calculate the delta ratio between the depletion and the saturation for both sensors, the minimum response (depletion) was obtained during the application of 10 mM pyruvate and the maximum response (saturation) during 10 mM lactate stimulation, their difference yielding the delta ratio.

To inhibit the different MCT isoforms expressed in the retina, we used a cocktail of 3 different drugs: SR-13800 0,1 μM (MCT1 blocker, Cat. No. 5431, Tocris, Bristol, UK), AR-C155858 2 μM (MCT1/MCT2 blocker, Cat. No. 4960, Tocris, Bristol, UK), and Syrosingopine 4 μM (MCT1/MCT4 blocker, Cat. No. SML1908, Sigma Aldrich, St. Louis, MO, USA).

### 4.7. Retinal Slice Preparation for FRET Experiments

For FRET experiments, the explants were separated from the culture inserts and placed in a chamber with extracellular solution, containing (in mM): 119 NaCl, 23 NaHCO_3_, 1.25 NaH_2_PO_4_, 2.5 KCl, 2.5 CaCl_2_, 1.5 MgSO_4_, 20 glucose, and 2 Na^+^ pyruvate, aerated with 95% O^2^ and 5% CO^2^, pH 7.4. The tissue was embedded in type VII agarose (Cat. No. 39346-81-1 Sigma-Aldrich) dissolved in a solution composed of (in mM): 119 NaCl, 25 HEPES, 1.25 NaH_2_PO_4_, 2.5 KCl, 2.5 CaCl_2_, 1.5 MgSO_4_ at a pH of 7.4. Subsequently, the tissue was cut with a vibratome (Leica VT1000S) to 200 μm thickness. The slices were transferred to the microscope recording chamber, sustained by a U-shaped platinum wire, and superfused with oxygenated extracellular solution at room temperature.

For FRET imaging, experiments were performed in an extracellular solution containing (in mM): 119 NaCl, 23 NaHCO_3_, 1.25 NaH_2_PO_4_, 2.5 KCl, 2.5 CaCl_2_, 1.5 MgSO_4_, 5 glucose, and 1 lactate, and aerated with 95% O_2_ and 5% CO_2_, pH 7.4. The lactate and glucose concentrations were chosen to not saturate the FRET sensors.

### 4.8. Photobiomodulation

Retinal explant cultures under moderate glucose (15 mM) or high glucose (30 mM) were treated with photobiomodulation (PBM) in an otherwise dark incubator. For this, we used a black box printed on a 3-day printer with two commercial LED devices inside, emitting at two different wavelengths: 525 nm (green, control) and 660 nm (NIR light), and the stimulation was set at 180 µW/cm^2^ for both wavelengths. PBM was applied for 24 h. continuously for 4 days, starting at p11, and at p15, organotypic retinal explants were fixed for subsequent analysis.

### 4.9. Data Analysis

Statistical analysis was performed using GraphPad Prism 8.0.1 software (RRID:SCR_002798). All data were first analyzed for normality using the Shapiro–Wilk test. If the test did not confirm a normal distribution, significant differences were established with the Kruskal–Wallis test followed by Dunn’s multiple comparison test. If the Shapiro–Wilk test determined a normal distribution, significant differences were established with the paired or unpaired *t*-test, whichever corresponded, and a one-way ANOVA multiple comparison, followed by Tukey’s multiple comparisons test. The α value was set to 0.05. Unless otherwise stated, data values are given as the mean± SD. Significance levels as indicated by asterisks were: * *p* < 0.05; ** *p* < 0.01, *** *p* < 0.001. Note that in many instances (Figure 3), the data shown are normalized versus control conditions, and the *p*-value indicates significance compared to the baseline or control.

## Figures and Tables

**Figure 1 ijms-24-07918-f001:**
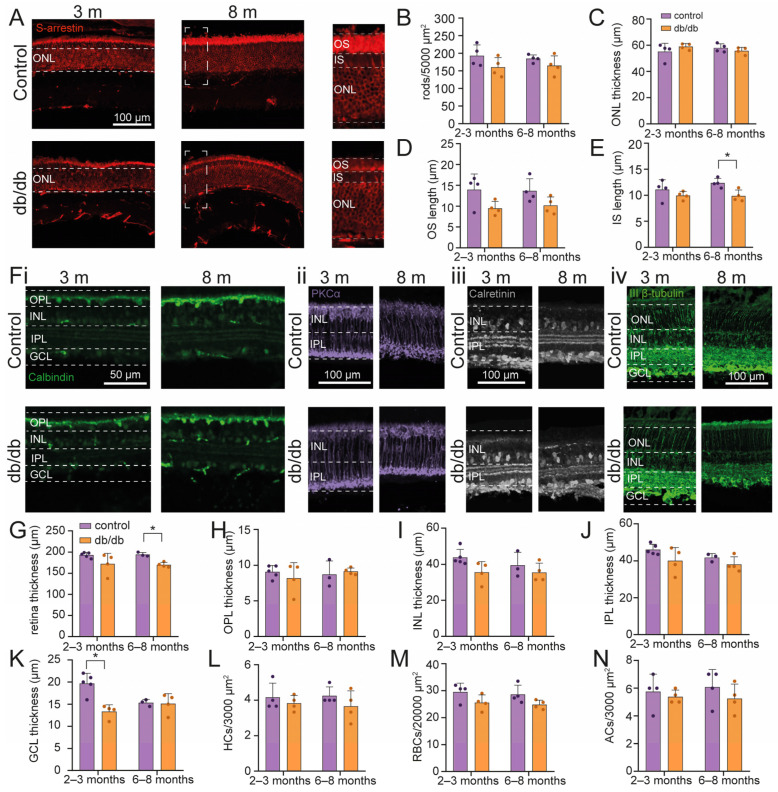
Analysis of different retinal cell types in diabetic db/db mice. (**A**,**F**) Representative immunofluorescence images for cell type-specific analysis: (**A**) Rod photoreceptors (s-arrestin), (**Fi**) horizontal cells (calbindin), (**Fii**) rod bipolar cells (PKCα), (**Fiii**) amacrine cells (calretinin), and (**Fiv**) ganglion cells (III β-tubulin). The scale bars are applicable to all images of the set. (**B**–**E**) Statistical analysis of different dimensional parameters of the outer retina, comparing db/db mice versus wild-type controls in young-adult versus old-adult mice. (**G**–**N**) Statistical analysis of total retinal thickness (**G**), outer plexiform layer (**H**), inner nuclear layer (**I**), inner plexiform layer (**J**), and ganglion cell layer thickness (**K**) in control and diabetic mice. (**L**–**N**) Statistical analysis of the density of horizontal cells (**L**), rod bipolar cells (**M**), and amacrine cells (**N**) in control and diabetic mice. Graph bars display mean± SD. Individual samples are represented as purple dots for control and orange dots for diabetic mice. Asterisks indicate * *p* < 0.05. OS = outer segment; IS = inner segment; ONL = outer nuclear layer; IPL = inner plexiform layer; INL = inner nuclear layer; IPL = inner plexiform layer; GCL = ganglion cell layer; HCs = horizontal cells; RBCs = rod bipolar cells; ACs = amacrine cells.

**Figure 2 ijms-24-07918-f002:**
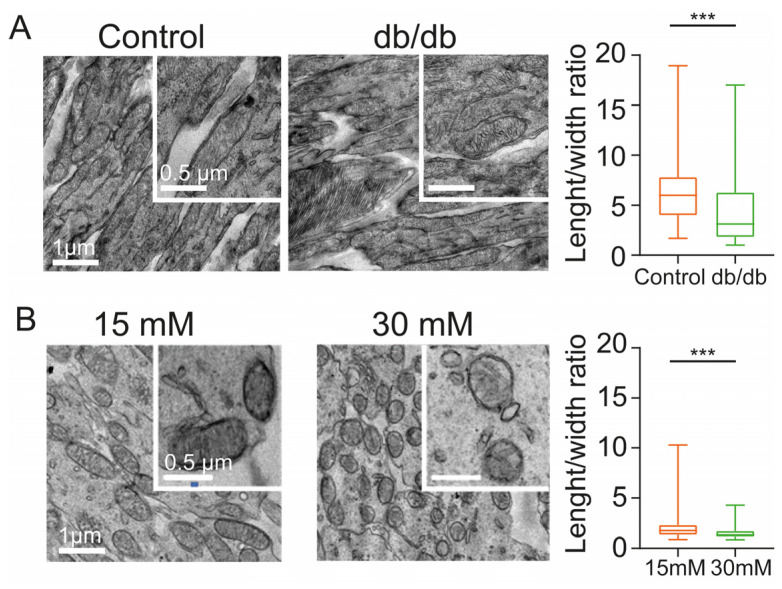
Ultrastructural analysis of photoreceptor inner segments shows alterations in mitochondrial morphology. ((**A**) left) Images of mitochondria located in the inner segments of photoreceptors in control and diabetic mice. ((**A**) right) Statistical analysis of length/width ratio of mitochondria in control and db/db (diabetic) mice. ((**B**) left) Images of mitochondria located in the inner segments of photoreceptors of retinal explants exposed to moderate glucose (15 mM) and high glucose (30 mM). ((**B**) right) Statistical analysis of length/width ratio of mitochondria in retinal explants cultured under 15 mM and 30 mM glucose concentration. The scale bars are applicable to all images of the set. Bar graph displays mean ± SD. Asterisks indicate *** *p* < 0.001.

**Figure 3 ijms-24-07918-f003:**
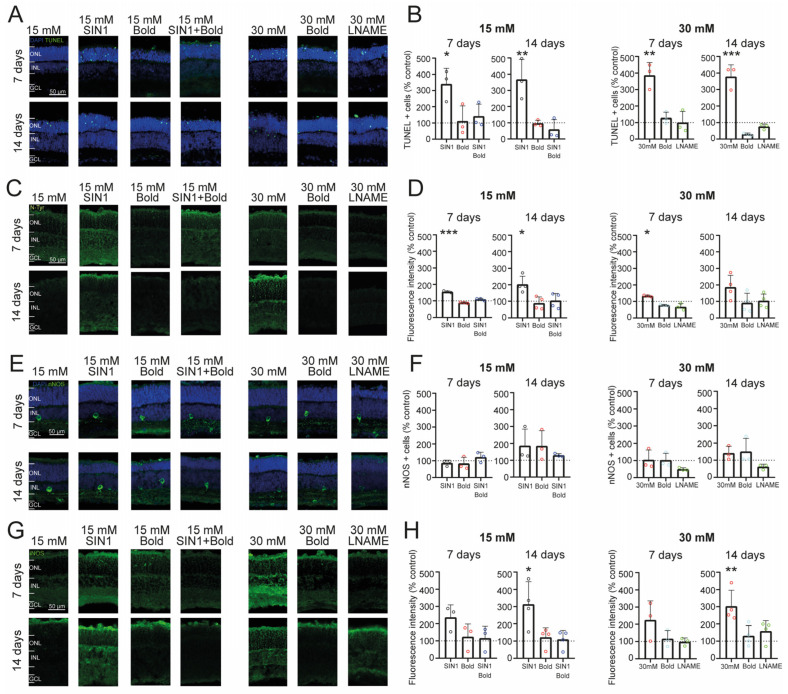
Expression of different nitrosative stress markers in organotypic retinal explants under different treatments. (**A**,**C**,**E**,**G**) Representative images of TUNEL (**A**), N-Tyr (**C**), nNOS (**E**), and iNOS (**G**) expression in retinal explants cultured for up two weeks under moderate glucose (15 mM) and high glucose (30 mM) conditions after different treatments. (**B**,**D**,**F**,**H**) Statistical analysis for moderate glucose (left), and high glucose conditions (right). Retinal explants cultured under high glucose displayed a high cell death rate (**B**), N-Tyr (**D**), and iNOS (**H**) expression, indicating that 30 mM glucose produces alteration in the nitrosative stress levels. These alterations are prevented after treatment with the antioxidant agent boldine, and the NOS inhibitor, L-NAME. All experiments are normalized to control (explants cultured with 15 mM glucose). The control is represented as a dashed line at 100%, and the results are shown as percentage of control. The scale bars are applicable to all images of the respective set. Graph bars display the mean ± SD, and individual explants are represented as circles. Asterisks indicate * *p* < 0.05, ** *p* < 0.01, *** *p* < 0.001. ONL = outer nuclear layer; INL = inner nuclear layer; GCL = ganglion cell layer; SIN-1 = peroxynitrite generator; bold = boldine; L-NAME = NOS inhibitor; N-Tyr = nitrotyrosine; nNOS = neuronal nitric oxide synthetase; iNOS = inducible nitric oxide synthetase.

**Figure 4 ijms-24-07918-f004:**
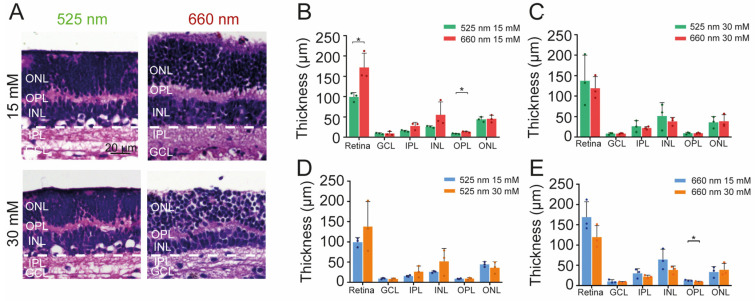
Application of PBM to retinal explants under high glucose. (**A**) Representative histological cross-section of retinal explants cultured under moderate glucose (15 mM, top) and high glucose (30 mM, bottom) treated either with 525 nm (left) or 660 nm light (right). (**B**–**E**) Statistical analysis of the thickness of retinal layers comparing different conditions: (**B**) moderate glucose and (**C**) high glucose treated with 525 nm vs 660 nm light; (**D**) effect of 525 nm and (**E**) 660 nm light stimulation in moderate vs high glucose conditions. The scale bar is valid for all images. Graph bars display the mean ± SD, and individual explants are represented as circles. Asterisks indicate * *p* < 0.05. Retina = total retinal thickness; ONL = outer nuclear layer; IPL = inner plexiform layer; INL = inner nuclear layer; IPL = inner plexiform layer; GCL = ganglion cell layer.

**Figure 5 ijms-24-07918-f005:**
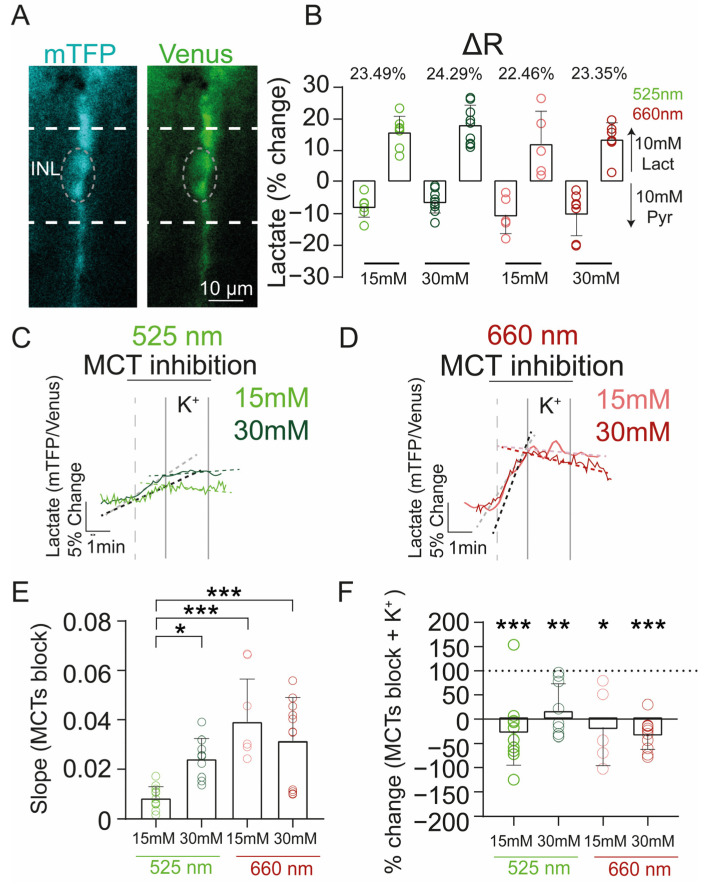
Lactate metabolism of MCs in retinal explants treated with PBM. (**A**) Representative images of the expression of the lactate sensor (Laconic). The scale bar is valid for both images. (**B**) Dynamic range of the lactate and glucose sensors in MCs showing no alterations due to light stimulation or glucose conditions. (**C**,**D**) Effects on lactate dynamics after inhibition of MCTs and retinal depolarization. Accumulation of intracellular lactate, as indicated by the steeply rising slope, was observed after bath application of MCT inhibitors in both glucose conditions. Black and grey dashed lines indicate drug-induced change in slope and red, green, light red, and light green dashed lines indicate the effect of drug + KCl. (**E**) Statistical analysis of the slope of the responses in the basal condition (drug only, left) and (**F**) under depolarization (drug + K^+^, right). There was an increase in lactate production between moderate and high glucose conditions under 525 nm treatment, and 660 nm treatment. The MCT-induced lactate increases diminished after depolarization under both glucose conditions and light conditions (525 nm 15 mM, *p* = 0.0007; 525 nm 30 mM, *p* = 0.0029; 660 nm 15 mM, *p* = 0.0362; and 660 nm 30 mM, *p* < 0.0001). The slopes were calculated by fitting a linear regression via the lineal function: y = mx + b. Data were analyzed with the paired Student’s *t*-test or one-way ANOVA, whichever corresponded. Graphs display the mean± SD. * Indicates *p* < 0.05; ** indicates *p* < 0.01; *** indicates *p* < 0.001.

**Figure 6 ijms-24-07918-f006:**
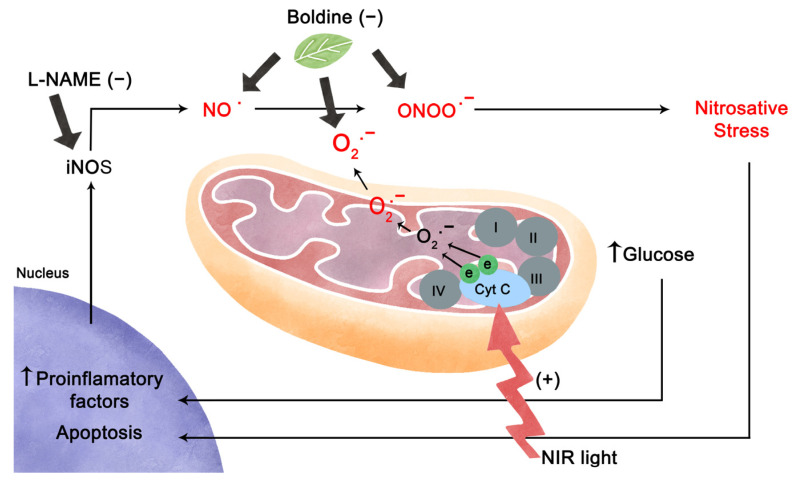
Scheme illustrating the putative relationship between high glucose, mitochondrial metabolism, oxidative/nitrosative stress, and the effects of PBM with NIR light. A high glucose condition (e.g., uncontrolled diabetes) disrupts mitochondrial morphology and functions, specifically affecting mitochondrial CCO, which increases anion superoxide levels (red). High glucose levels also increase proinflammatory factors and cause an upregulation of iNOS, leading to pathologically high NO concentrations. NO reacts with the anion superoxide to produce peroxynitrite, thus raising the oxidative stress levels in cells (red), triggering apoptosis. In this scenario, L-NAME acting on iNOS regulates NO levels together with boldine, which is an antioxidant that reduces NO, anion superoxide, and peroxynitrite levels. On the other hand, PBM enhances mitochondrial metabolism (red flash to mitochondria), improving CCO activity, and reducing the liberation of free radicals. NIR = near-infrared light; iNOS = inducible nitric oxide synthetase; NO = nitric oxide; CCO = cytochrome c oxidase; Cyt C = cytochrome c.

## Data Availability

The data presented in this study are available on request from the corresponding author.

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
