# Peer review of "Evaluation of Photobiomodulation and Boldine as Alternative Treatment Options in Two Diabetic Retinopathy Models"

_ijms, 2023, doi:10.3390/ijms24097918_

Round 1

Reviewer 1 Report

The authors study the effect of antioxidant, NOS inhibitor and photobiomodulation treatment on the retinal explants under high glucose conditions. The beneficial effect of antioxidant and NOS inhibitor has been well studied and supported by the results. The effect of PBM on diabetic retinopathy is not clear from the results presented in this study. The study can be improved by addressing the following concerns,

1. The title is a bit long. It is not necessary to list the types of analysis performed in the title. A shorter title is recommended.

2. In the introduction, the description of previous studies on the effect of PBM in DR models is very limited in lines 61-63. It must be expanded to describe the findings from previous studies in better detail and compare  with the goal of the current study.

3. In figure 2, the individual data points must be shown in the bar graphs similar to the figure 1. 

4. Figure 3 and the corresponding description in the results can be improved. The immunofluorescence images and the corresponding quantifications should be shown together. For example, 3C should be moved to 3B as it shows the quantification of the images from 3A. The description in the results section can be simplified for Fig 3.

5. The quantification of TUNEL staining in Fig 3C does not correlate well with the immunofluorescence images in Fig 3A. For instance the 7d SIN1+Bold image for 15 mM glucose has equivalent or more TUNEL staining than the 7d SIN1 image. The color label for the stains and DAPI is not very clear in the figures.

6. The effect of PBM on retinal explants under high glucose condition is similar to that of moderate glucose condition. So, it is unclear whether PBM would have beneficial effect on diabetic retinopathy. If it is proposed as a complementary option to Boldine, then the effect of PBM and Boldine combination should have been studied. 

7. Does PBM affect the cell death rate in retinal explants under high glucose condition?

Author Response

Dear reviewer, thank you for your valuable comments on our manuscript. We have made our best efforts to respond to all of your suggestions as accurately as possible:

1)  The tittle has been shortened to be more precise and concise, now mentioning both PBM and boldine as interventions in the study.

2) We improved the figures. In Figure 2, we modified the graph displaying the data as box and whiskers to show the results more accurately. In addition, Figure 3 was modified as recommended moving all quantification to the right, and the TUNEL staining was updated to correlate the graph results with the fluorescence images.

3) Regarding the possible effects of PBM on the cell death rate in retinal explants cultures, we agree that is an experiment should be done and will be included in a subsequent paper, as this one is already quite voluminous.

4) Finally, we completed Figure 6, showing results of mitochondrial oxidation in retinal explants treated with both 525 nm and 660 nm LED light, improving the conclusion of our study.

Best regards

Victor Calbiague and Oliver Schmachtenberg

Reviewer 2 Report

The authors, in two diabetic model systems, tested the potential benefits of photobiomodulation as a non-invasive treatment option for diabetic retinopathy, with the results suggesting that while photobiomodulation improved mitochondrial metabolism, it was insufficient to improve retinal structural damage caused by high glucose conditions, warranting further research into its potential as a complementary treatment option. The authors should be praised for their hard work, however, several aspects of the manuscript require extensive restructuring and clarification.

The paper is somewhat challenging to comprehend as it does not provide clear information regarding the number of mice and controls analyzed. Further, you should more clearly specify that you are evaluating the efficacy of PMD in an in vitro setting only. While I will provide some initial comments, I may need to include additional feedback after further revision (if it will be proposed by the editor), as the article's lack of clarity makes it challenging to offer substantive feedback.

General comment: I am uncertain whether an ophthalmologist is among the authors, as some sentences regarding the clinical management of diabetic retinopathy appear imprecise, and if this is not the case, it would be beneficial to have an ophthalmologist review the manuscript to enhance certain sections.

Abstract:

1) line 14: "Current treatment are invasive and only applied at advanced stages". This is partially accurate as while glucose control is considered the primary treatment for diabetic retinopathy, anti-VEGF agents (and in some cases steroids) and laser therapy may also be employed during the early stages of the condition, especially when signs of ischemia or cystoid macular edema are observed. However, the fact that treatments are more commonly administered at advanced stages, can be attributed to the lack of preventive strategies. Moreover I would define intravitreal injection as "quite invasive".

2) lines 15-16: "Photobiomodulation [...] has shown promising results in ameliorating retinal pathologies and insults". Again, this statement is partially true. It should be clearly stated in the abstract that the efficacy of PBM is a subject of controversy and debate.

3) line 18. The final number of the studies retinal explants as well as the number of involved mice should be clearly stated in the abstract.

Introduction: The introduction should primarily concentrate on the literature surrounding the relationship between PBM and diabetic retinopathy (DR), rather than solely on the efficacy of PBM in treating ocular conditions other than DR.

4) line 32. Replace "retinal detachment" with "Retinal traction detachment". Add also "ischemia" and "retinal hemorragies". 

5) lines 36-38. The authors should provide a clearer distinction between the two types of diabetic retinopathy, namely proliferative and non-proliferative diabetic retinopathy, given their distinct pathological features.

6) lines 38-42. I do not understand why five lines were devoted to ERG findings in non-recent articles primarily involving animal models. While ERG is a method for early detection of diabetic retinopathy, it is likely not the most frequently used approach in clinical practice. The authors should provide more extensive references to other methods such as OCT and OCT-angiography, fluorescein angiography, or visual field tests.

7) lines 42-43. Replace "are directed at 42 the vascular alterations" with "are primarily directed". Moreover "only applied to advanced stages" is partially true as already discussed (see point 1).

8) line 44. Treatments involve also laser photocoagulation and vitrectomy. Also, intravitreal injections are not painful. And again I would define intravitreal injections as "quite invasive".

9) lines 48-60. This paragraph should be rewritten referring to the most recent evidence about PBM and diabetic retinopathy. 

-"Since the retina is difficult to target pharmacologically due to the blood-retina barrier" - this is partially true: what about intravitreal injections? It is indeed a very easy way to target the retina.

-"Moderately positive effects have been shown in retinopathy of prematurity (ROP)". This is not true. Whereas Natoli and al. reports some results, more recently Kent et al did not find any real benefit.

-The paragraph needs to be revised to highlight that there is currently insufficient evidence to support the efficacy of PBM. In this regard, it would be appropriate to reference the only available randomized trial examining the effectiveness of PBM in treating diabetic retinopathy, which was conducted by Kim et al. (doi:10.1016/j.oret.2021.10.003), and found no evidence of efficacy. The paragraph should strive for greater impartiality.

-lines 57-60 are not particularly relevant to the article's topic. Use this space to discuss more recent articles in greater depth.

10) line 61. "PBM has been tested in few studies". Which ones? Please clearly specify in the previous paragraph.

11) lines 61-63: "which have found [...] may improve". This is generally inaccurate, as there is currently a lack of strong evidence available. By explicitly stating the lack of effectiveness in the literature, the subsequent sentence "here we set out" will make more sense.

Result: The Results section is difficult to comprehend for readers. I would recommend that the Results section only include raw data, and avoid adding elements that should be incorporated into the Methods or Discussion sections instead. Moreover, authors should try to avoid the expression "statistically significant" as it has been recently criticized (https://www.nature.com/articles/d41586-019-00857-9). They could use "we found a difference between" and "we found no differences between". Or yet, "we found high compatibility" or "low compatibility". 

General comment: you should start the results section clearly stating how many mice and retinal explants you analyzed. You should help the reader to immediately understand the amount and the relevance of your work.

12) Lines 73-78 should be partially incorporated in the Methods section.

13) Lines 79-80 how many 3-months mice and 8-months mice you have analyzed? And how manu control retinas at 3-months and 8-months you have evaluated? Probably - I imagine - 3 or 4 per group looking at the figures. But you should definitely and clearly and state that in the results section and it should be also clear in the Figure 1.

14) Line 81 "length of the inner segment". It makes no sense. You are probably talking about "thickness".

15) Line 82 what is the meaning of "remained visibly unaltered"? You visibly evaluated the retina or you measured it?

16) Lines 76-86 please add p-values and/or even better confidence intervals (CI). The term "non-significant reduction" lacks statistical significance. If the result is deemed "non-significant," any minimal differences should not be referred to as a "reduction." Moreover if you are comparing OS between diabetic mice and control mice and IS between diabetic mice and control mice, why are you reporting OS and IS together in same parentheses? It would be more comprehensible to report OS values from control mice and diabetic mice at 8 months in the same parentheses. The same of IS and for the 3-months mice.

17) Line 88: "diameter of retinas". It makes no sense. You are probably talking about "thickness".

18) Line 91: "Small but non-significant reduction". Avoid this type of expression. 

19) Lines 109-114: This part should be moved to the Discussion section. Additionally, while you observed an overall reduction in retinal thickness in 8-month-old mice, there were no differences found in the individual retinal layers. How do you explain this finding, considering that the overall retinal thickness is the sum of the thicknesses of each individual layer? This observation is puzzling and requires further explanation.

20) Lines 130-139. This part should be moved to the introduction and/or the discussion sections.

21) Lines 150-151 "Previously etc". This sentence should be moved elsewhere (method? discussion?).

22) Lines 175-182. This part should be move in the methods and/or discussion sections. The same for lines 185-187.

23) Line 190 "Note that the p-value indicates significance compared to controls". That sounds a bit of out context. Is there another way to specify this in the previous sentence?

24) Lines 203-205 cannot be part of the result section, they should be moved in the discussion section.

25) Lines 195-239. I am a bit confused as to how these experiments are relevant to the aims of the study. While I understand that NO plays a role in cell death, boldine acts as an antioxidant, and high glucose creates a nitrosative environment in the retina, I fail to see how these findings align with the stated objectives of the study. Perhaps these experiments deserve to be included in a separate study? Maybe you have enough material to publish two articles? Or you probably declare your aims more clearly at the end of the introduction section.

26) Lines 255-288. Similar doubts about the relevance of this findings in studying the efficacy of PBM in diabetic retinopathy.

27) Lines 291-297 should be put elsewhere (not in the results section).

28) Line 298 The number of evaluated explants should be clearly stated.

29) Lines 302-303. The statement that "PBM treatment with NIR light caused a significant increase in overall retinal thickness under moderate glucose conditions" is difficult to comprehend. This finding seems implausible, as the retina does not typically grow or thicken like a cake in response to external stimuli. There is currently no evidence indicating that NIR can cause a significant increase in retinal thickness, which raises more than one question. Further explanation will be necessary in the Discussion section.

30) 322-334 lines. This part should be discussed in the Methods section. You can leave a brief summary (one sentence) here.

31) 336-339 lines. "The results are in line". This is part of the Discussion section.

32) 342-346. Put this part in the methods section.

33) line 354-360 I would put this is part in the discussion

Discussion: would recommend revising this section to include all the elements that were previously presented in the Results section. Additionally, the Discussion seems to be primarily centered on the results related to the antioxidant Boldine. This is somewhat confusing, as Boldine was not specifically mentioned in the aims or the introduction of the article. The reader may not realize until the end of the Discussion section that Boldine was also a focus of investigation. As such, it would be helpful to provide more clarity and context around the inclusion of Boldine in the study. Probably it worths to be moved in a distinct article?

33) Line 376 To aid readers in comprehending the research, I recommend summarizing the Discussion section with one or two sentences that highlight the main findings of the study.

34) Lines 401-402. Replace "the development of diabetic retinopathy" with "oxidative stress and local inflammation in diabetic retinopathy".

35) You have to clearly mention the strengths and especially the shortcomings of your study. E.g., the small sample size, the in vitro setting of the experiments etc.

36) You have to propose future research and methods to apply to obtain better and stronger results 

Methods I suggest improving this section by incorporating the methodological details that were previously presented in the Results section..

37) line 516. How many animals you included in this studied? How many animals were sacrificed? Please clearly state the number of employed mice in the research.

38) line 586. Please move in the Data Analysis paragraph

Author Response

Dear Reviewer,

thank you for your thorough and highly knowledgeable revision of our manuscript.

We have tried to remedy almost all of the shortcomings you have depicted in our manuscript, and have changed the “tone” to one more balanced regarding the available evidence for photobiomodulation efficacy.

1) First, we have modified the introducction, including a detailed section on the recent literature about PBM application on diabetic retinopathy models. Parts less relevant for the study were removed. Furthermore, we changed and improved the decripton of all concepts, hoping that they are now more accurate from a clinical point of view.

2) In the results sections, we have stated how many mice and retinal explants were analyzed in the different experiments. We also added the p values of each results, to show the specific values of our statistics. We deleted the expression “statistically significant” in accordance with recent recommendations. Regarding the parts recommended to be moved from this section, we rewrote most of the phrases mentioned, and we moved parts to the respective recommended sections. We also added an experiment in Figure 6, showing the effects of 525 nm LED light treatment on mitochondrial oxidation in retinal explant culture. Hence, we complete the data on the effects  of both wavelength conditions on lactate metabolism.  

3) The discussion was modified as recommended. In the beginning of the discussion we added a brief summary of the main findings of the study. Also, we added an special section discussing the strengths and shortcommings of our study, focusing on the sample size, and stating the importance of future experiments to confirm the results observed in our in vitro condition.

4) Finally, we improved the methods section incorporating the methodological details. Specifically, we now indicate the total number of animals an explants used in this study in total and for each experiment, and we detail the drugs used in the description of  “FRET measurements”.

We hope that these changes will improve the manuscript significanlty in your opinion and make it acceptable for publication in the International Journal of Molecular Sciences.

Best regards

Victor Calbiague and Oliver Schmachtenberg

Round 2

Reviewer 1 Report

The authors have addressed most of the concerns in the revised manuscript.

Author Response

Many thanks for your comments, which have helped us improve the manuscript.

Reviewer 2 Report

The authors revised the manuscript according to my suggestions; however, they did not respond point-by-point to my raised questions. 

The authors have clearly improved the manuscript; however, some points are still worth of attention. To evaluate your revision more quickly, please answer below each of my raised points.

Abstract:

1)    Lines 16-17. The authors state: “The Boldine is a potent natural antioxidant and potentially useful to prevent free 16 radical-induced oxidative stress”. This sentence here seems out of context. Please try to rephrase.

2)    Line 19. Are the 97 retinal explants coming from the wild type mice or from both wild type and obese/diabetic mice?

3)    Reading the abstract, it is not clear if you analyzed the retinal explants also from the obese/diabetic mice. You should probably rephrase. You specify the number of obese/diabetic animals but not the number of wild-type animals.

4)    If you state in the abstract that you analyzed 97 retinal explants, it seems that you analyzed the retinal explants altogether. Reading the manuscript, I understand that you employed a TOTAL of 97 retinal explants to conduct a series of experiments. And this is totally different. Please state clearly state that every experiment was not conducted on 97 retinal explants, but only on a little a part of them (e.g., 3-5 for each experiment).

Results:

5)    110. Have you conducted the experiments on retinal explants? Please specify.

6)    Lines 113-114. You state that you analyzed the effects of baseline diabetic condition on the retina of 8 animals (4 diabetics and 4 controls). Since mice have two eyes, it could be up to 16 retinal explants. Does the number of animals coincide with the number of examined retinal explants?

7)    Lines 117-125. If the number of animals is the same in all those experiments, probably no need to always specify the number of animals. One time at the beginning is ok.

8)    Line 123. P-value has a comma instead of a point.

9)    Line 126-131. Why does the number of controls and db/db mice change at every experiment? You should explain why you are using different number of retinal explants (or animals) when testing the retinal thickness.

10) Lines 150-154. This part should be moved in the discussion section. And please avoid expressions like: “most significant reduction“.

11) Lines 170-176. This part should be moved elsewhere (Introduction? Discussion? Methods?). I suggest you define a little protocol in the methods section. Better not to state in the results that "since you found a particular thing, then you decided to do another thing". Methods should be defined before the results. Better not to change or define methods according to results.

12) Lines 185-187. What n=138 and n=135 refer to? I understand that the number of animals is: 3 wild-type and 3 db/db. Please explain.

13) Lines 188-189. Ok. But this should be placed in the Introduction or in the Discussion.

14) Lines 195-198. What n=626 and n=606 refer to? The number of animals is still 3 and 3. Please explain.

15) Lines 199-201. I would put this part in the discussion section.

16) Line 218. How many of the 42 retinal explants were wild type and how many db/db?

17) Lines 223-230. Please specify the values of the control conditions. I cannot reproduce your analyses.

18) Lines 231-234. As for lines 170-176, I would move these explanations elsewhere.

19) Lines 239-241. I would move it to the discussion section.

20) Line 271. P-value is 0.9998. Why do you state that nitrosative stress decreased at 14d?

21) Lines 261-271. Please state the values of controls to reproduce the results.

22) Line 301. Same as Line 218.

23) Lines 297-305. These explanations cannot be part of the results section, as they refer to methodological procedures. You should not explain here why you choose to do a procedure instead of another. You should explain it in the methods section.

24) Lines 310-319. Please specify the values of the controls to reproduce the results.

25) Line 336. Add “A total of 24”

26) Lines 336-337. How many db/db, how many controls?

27) Line 331 (all the paragraph). As stated before, please move comments or explanations about the experiments in other sections. Please add the values of the controls for each result. Or better, you can report them on a table.

Discussion:

Discussion section has been improved.

Materials and Methods:

28)  Line 576. You state that you raised 61 Wild type C57Bl/6 mice. Then, at line 585 you state that retinal explants were obtained from 50 P9 wild-type mice. Are they 61 or 50? Please explain.

29) It is not clear how the obese/diabetic mice were analyzed. Reading the methods it seems that the retinal explants came only from the wild type mice. And I don't think so. How many retinal explants you took from the obese/diabetic mice?

Author Response

Dear Reviewer,

Thank you for your valuable comments on our manuscript. we have made our best efforts to respond to them, as accurately as possible:

1) Point 1. First, we have extensively reformulated the abstract, to better reflect the study and make it clearer.

Points 2, 3, and 4: The origin and number of the organotypic retinal explants and their experimental distribution, as well as the numbers of wild-type and transgenic mice, are now clearly stated in the methods and results sections. However, we finally decided against adding animal and experiment numbers to the abstract, which is not a very common practice, and since they are included elsewhere.  

2) The result section was also modified:

Points 5, 6, 7, 10, 12, 14, 16, 22, 25, 26, and 27 were modified as recommended.

Point 5: Retinal explants were obtained only from wild-type retinas, but not from diabetic mice.

Point 6: We explained that the experiments in Figure 1 were run in ex vivo retinas but not in explants.

Point 10: We deleted the expression “the most significant reduction.”

Points 12 and 14: We have clarified that N = number of animals and n = number of mitochondria.

Points 16, 22, and 26: We explained in Point 5 that retinal explant cultures were obtained only from wild-type retinas.

Point 25: We have added the phrase recommended on line 336.

Point 27: We have added the values of the controls to the explanation of each result in lines 347-350.

Regarding the points 11, 13, 15, 18, 19, and 23:

We do not believe that these phrases should be moved to any section. It is important to add some connecting phrases and to introduce some definitions. We understand that the lecture of a scientific article should have a certain order (introduction, methods, results, discussion); however, this does rarely happen, and many times readers jump straight to specific parts. To make the text easier to read, these phrases should be placed there. In addition, finishing the different results sections with a small conclusion is helpful for summarizing and understanding each main result.

3) Finally, we improved the Methods section, indicating how many animals were used in each experiment to clarify any doubts. In the investigation, we used 61 wild type animals divided into 8 animals for immunofluorescence, 3 animals for electron microscopy and 50 animals for retinal explant cultures experiments. Also, we indicated that 11 diabetic mice were used, divided into: 8 animals for immunofluorescence and 3 animals for electron microscopy.

We hope that these changes will improve the manuscript significanlty in your opinion and make it acceptable for publication in the International Journal of Molecular Sciences.

Best regards

Victor Calbiague and Oliver Schmachtenberg

Round 3

Reviewer 2 Report

The authors have provided thorough responses to my inquiries, which have helped to clarify the previously ambiguous aspects. Perhaps the inclusion of one or more tables could have made the results section more concise. Although we may have differing views on placing the explanation of methodological procedures in the results section, I believe that the manuscript has significantly improved since its initial submission, and I commend the authors for their hard work.